



# Technical Note: Evaluation of a low-cost evaporation protection method for portable water samplers

Jana von Freyberg[1,2,3,*], Julia L. A. Knapp[1,*], Andrea Rücker[1], Bjørn Studer[1], and James W. Kirchner[1,2,4]

[1]Department of Environmental Systems Science, ETHZ, 8092 Zurich, Switzerland
[2]Mountain Hydrology and Mass Movements, Swiss Federal Institute for Forest, Snow and Landscape Research (WSL), 8903 Birmensdorf, Switzerland
[3]School of Architecture, Civil and Environmental Engineering, EPFL, 1015 Lausanne, Switzerland
[4]Department of Earth and Planetary Science, University of California, Berkeley, CA 94720, USA
[*]both authors contributed equally

*Correspondence to*:
   Jana von Freyberg (jana.vonfreyberg@epfl.ch)
   Julia L. A. Knapp (julia.knapp@usys.ethz.ch)

**Abstract** Automated field sampling of streamwater or precipitation for subsequent analysis of stable water isotopes ($^2$H and $^{18}$O) is often conducted with off-the-shelf automated samplers. However, water samples stored in the field for days and weeks in open bottles inside autosamplers undergo isotopic fractionation and vapor mixing, thus altering their isotopic signature. We therefore designed an evaporation protection method which modifies autosampler bottles using a syringe housing and silicone tube, and tested whether this method reduces evaporative fractionation and vapor mixing in water samples stored for up to 24 days in ISCO autosamplers (Teledyne ISCO., Lincoln, US). Laboratory and field tests under different temperature and humidity conditions showed that water samples in bottles with evaporation protection were far less altered by evaporative fractionation and vapor mixing than samples in conventional open bottles. Our design is a cost-efficient approach to upgrade the 1-litre sample bottles of ISCO 6712 Full-size Portable Samplers, allowing secure water sample collection in warm and dry environments. Our design can be readily adapted (e.g., by using a different syringe size) to fit the bottles used by many other field autosamplers.

## 1 Introduction

The stable water isotopes deuterium ($^2$H) and oxygen-18 ($^{18}$O) are used as natural tracers for water flow through the landscape, and thus provide important insights into water sources, flowpaths, and travel times in hydrologic systems (e.g., Gat et al., 2001; Kendall and McDonnell, 1998; Klaus and McDonnell, 2013; McGuire and McDonnell, 2008). Furthermore, deuterium and oxygen-18 signatures in precipitation and/or streamwater can help to track the movement of atmospheric air masses (Fischer et al., 2017), identify the water sources of plants (Dawson and Ehleringer, 1991), and reconstruct climate





records (Shanley et al., 1998). Long-term data sets of stable water isotopes in precipitation and streamwater are available from global monitoring networks (the Global Network of Isotopes in Precipitation, GNIP, and the Global Network of

Isotopes in Rivers, GNIR) and various national monitoring networks (e.g., the ISOT monitoring program of the Swiss Federal Office for the Environment).

Streamwater is usually collected through instantaneous grab sampling, after which the sample containers are sealed and cooled until laboratory analysis. In contrast, precipitation is usually collected over periods of weeks to months with open

buckets or funnels mounted onto sample bottles. To prevent evaporative fractionation of the precipitation sample during the sampling period, paraffin oil can be used to form a layer of oil floating on the water sample, thus preventing evaporative fractionation (IAEA, 2014). However, residual oil in the water sample can alter subsequent laser spectroscopy measurements (Gröning et al., 2012). The contamination risk is particularly high if the sample volume is small, so the addition of oil is only suitable for longer sampling durations (weekly or monthly), but not recommended for daily or sub-

daily sampling. Alternative mechanical evaporation protection modifications have been suggested, like placing a table tennis ball in the collection funnel ("ball-in-funnel") to seal the inflow during times without precipitation (Prechsl et al., 2014). Another widely used collector modification is the "tube-dip-in-water" collector (Gröning et al., 2012; IAEA, 2002), where the collection bottle is sealed except for a small-diameter tube that reaches from the bottom outlet of the funnel into the water sample. This setup substantially reduces the contact area between the sample and the atmosphere. While some of

these modifications may substantially reduce evaporative fractionation of the water sample in the bottle, others were found to be less effective (Michelsen et al., 2018; Terzer et al., 2016).

The above methods and modifications were originally designed for single-sample collection using a precipitation totalizer (e.g., IAEA, 2014). For many hydrological questions, however, higher-frequency measurements of stable water isotopes are

of interest, requiring daily or even sub-daily sampling of precipitation or streamwater (e.g., Knapp et al., 2019; Rücker et al., 2019; von Freyberg et al., 2018; Wang et al., 2019). This can be achieved with field-deployable automatic water samplers with programmable pump-and-distribution systems that fill and store a series of empty open bottles. Many hydrologic studies deploy off-the-shelf automatic water samplers (available from, e.g., Teledyne ISCO, Lincoln (NE), USA, and Maxx GmbH, Rangendingen, Germany), because these systems are rugged, robust, versatile, and easy to program. For automatic

samplers with a 24-bottle configuration, this setup reduces the manual labor of daily precipitation sampling to the collection of sample bottles only once every 24 days. However, because the sample bottles remain open during the sampling period, the water samples are exposed to changes in air temperature and humidity inside the autosampler housing, which may alter the isotopic composition of the water sample in the bottles due to evaporative fractionation and mixing in the vapor phase.

While attempts have been made to design more sophisticated field-deployable, programmable water samplers which reduce these isotope fractionation effects, most of these devices are not readily available (i.e., prototypes), or are technically





complex or expensive (Ankor et al., 2019; Berman et al., 2009; Hartmann et al., 2018; Michelsen et al., 2019). We therefore designed and tested a low-cost evaporation protection modification that can be used with Teledyne ISCO's 6712 Full-size Portable Samplers. We retrofitted the 1-litre ISCO sample bottles with a simplified "tube-dip-in-water" collector type that allows rapid sample flow, but reduces evaporative fractionation. The proposed setup is cheap, easy to handle and suitable for a wide range of sample volumes that are common in daily precipitation or streamwater sampling.

## 2 Methods

### 2.1 Evaporation protection

We designed an evaporation protection modification for the 1-litre sample bottles of the 6712 Full-size Portable Sampler (Teledyne ISCO., Lincoln, US; hereafter referred to as 'ISCO autosampler'). Our evaporation protection consists of a 100-ml syringe housing (i.e., BP Plastipak[TM] 100ml syringe with catheter tip, without its piston and rubber piston stopper) with attached Luer tip adapter (BP Plastipak[TM]). On the Luer tip, we fit a 1-mm inner diameter silicone tube approximately 9 cm in length, to reach the bottom of the sample bottle (Figure 1b). The barrel flange of the syringe housing is trimmed on one side (Figure 1a) to allow the retrofitted sample bottles to properly fit into the ISCO sampler ensuring that they do not block the distributor arm. This modified syringe housing is then plugged tightly into the opening of an ISCO sample bottle (Figure 1b). Our setup ensures a smooth, splash-free sample flow from the syringe through the silicone tube into the bottle. Because the end of the silicone tube is fully immersed in the sample liquid, only the cross-sectional area of the silicone tube is exposed to the ambient atmosphere (rather than the entire cross-sectional area of the water surface), minimizing vapor exchange with the surrounding atmosphere. Our design of the evaporation-protected ISCO bottle is robust, cheap (<5 USD/sample bottle), chemically inert, easy to disassemble and to clean, and allows rapid sample flow of approximately 100 ml min[-1] into the bottle.

In field operation for the collection of streamwater samples, the autosampler should be programmed to not exceed the filling rate that can be accommodated by the narrow silicone tube. This can be accomplished by programming the autosampler to deliver a series of 100 ml aliquots, allowing enough time between them (about 1 min minimum) so that they can drain from the 100 ml syringe into the sample bottle. If possible, one should also limit the total sample volume so that the water line is somewhere in the narrow silicone tube and not in the syringe, in order to limit the water surface that is available for evaporation or condensation. To transport the filled sample bottles, the syringe housing has to be removed and the bottles have to be sealed with screw caps supplied by the manufacturer. If the bottles are transported upright and leakage is unlikely to occur, the syringe housing can also stay in place and its upper opening can be sealed with the black rubber piston stoppers that are supplied together with the syringes (BP Plastipak[TM] 100 ml syringe with catheter tip; Figure 1c).



## 2.2 Monitoring evaporation and fractionation

We conducted three experiments to assess the effects of evaporation and vapor mixing on the isotopic composition of the liquid samples in ISCO 6712 autosamplers, comparing the retrofitted ISCO bottles to un-modified ISCO bottles. In
Experiment 1, we simulated a daily sampling routine under extremely dry and warm conditions to test for evaporative fractionation effects over different storage durations. In Experiment 2, we used two contrasting reference waters to test for changes in their isotopic compositions due to vapor transfer between samples, in addition to fractionation effects under ambient conditions with diurnal fluctuations in temperature and humidity. Experiment 3 evaluated the performance of the retrofitted ISCO bottles during 62 3-week cycles over a nearly 4-year deployment at two field sites in the northern Swiss pre-
Alps.

### Experiment 1

We prepared one ISCO autosampler for a 24-day test of the retrofitted bottles under controlled laboratory conditions. The autosampler contained 24 sample bottles, of which 12 were retrofitted with the modified syringe housing and the other
12 bottles remained open (i.e., as they do in normal operation). Open and retrofitted bottles were arranged alternatingly in the autosampler carousel. The ISCO autosampler was placed inside a climate-controlled chamber where the conditions were kept at approximately 35 °C air temperature and 11 % relative humidity. Air temperature and relative humidity were measured every hour in the climate chamber and inside the bottom compartment of the ISCO autosampler.

The bottom compartment (containing the sample bottles) and the middle compartment (containing the pump and control unit) of the ISCO autosampler remained attached for the entire duration of the experiment. The water samples were distributed among the bottles by using the instruments' software to move the distributor arm to the desired position. The instrument's sampling tube was not threaded through the peristaltic pump, but instead was directly attached to the inlet of the distributor arm. This setup allowed an exact amount of water to flow gravitationally from the inlet tube through the
distributor arm into the sample bottle (we bypassed the peristaltic pump because it does not allow such exact sample dosing and might introduce air bubbles into the sample during pumping). This sampling protocol is consistent with the automated sampling of precipitation under field conditions, when the autosampler's sample inlet tube is connected directly to a precipitation collection funnel (i.e., bypassing the peristaltic pump) so that incoming precipitation flows directly through the distributor arm towards the pre-programed bottle position (e.g., Rücker et al., 2018).


To ensure that the initial isotopic compositions of all water samples were comparable, we filled a 20-litre tank with distilled reference water before the beginning of the monitoring period. This reference water tank was tightly sealed and stored at room temperature. It was only opened every second day to retrieve 801.5 ml of reference water. From this aliquot, 1.5 ml were filled into a glass vial with screw cap (screw thread vials 1.5ml, PP-screw thread caps with silicone-/PTFE-septum,





WICOM Germany GmbH, Heppenheim, Germany) and stored at 4 °C until isotope analysis. The purpose of these samples was to monitor the isotopic composition of the reference water and account for possible fractionation inside the storage tank. The remaining 800 ml of reference water was filled into two empty ISCO sample bottles (400 ml each into an open and a retrofitted bottle). For the open bottle, 400 ml were emptied rapidly into the inlet tube. Because of the small tubing diameter in the retrofitted sample bottles, we poured the 400 ml of reference water into the inlet tube in four steps of 100 ml min$^{-1}$ to

prevent overflow.

Starting on day 1 and then every second day, one open and one retrofitted bottle were filled with 400 ml of reference water each following the protocol described above, and the last two sample bottles (No. 23 and 24) were filled at the start of the 23$^{rd}$ day.


To monitor evaporation and isotopic fractionation under ambient conditions outside the ISCO autosampler, we prepared three additional ISCO bottles at the start of the monitoring period. For this purpose, we filled 400 ml of the reference water each into one open ISCO bottle (i.e., non-modified), one ISCO bottle that was retrofitted with evaporation protection, and one tightly sealed ISCO bottle on day 1 of the laboratory experiment. We placed these bottles inside the climate-controlled

chamber, but outside the ISCO autosampler, for the duration of the experiment (24 days).

To mimic the field protocol (see Sect. 2.3), all sample bottles (i.e., inside and outside of the ISCO autosampler) were opened and sub-sampled at the end of day 24. For this, 1.5 ml of water from each sample bottle were immediately transferred into glass vials with screw caps and stored at 4°C until isotope analysis.


**Experiment 2**

For Experiment 2, we prepared two ISCO autosamplers with alternating open and retrofitted bottles, analogously to Experiment 1. One sampler was stored indoors at approximately constant temperature and relative humidity, and the other sampler was stored outdoors at a sunny location where ambient conditions were more variable. Temperature and relative

humidity were monitored inside and outside the ISCO autosamplers at both locations.

We filled all sample bottles on day 1 of the experiment to ensure that all samples underwent the same mixing and fractionation processes over the following 21 days. We alternatingly filled the bottle pairs (open and retrofitted) with two isotopically contrasting reference waters: one, which we will call RefA, was isotopically much heavier ($\delta^2$H ≈ -40.5 ‰,

$\delta^{18}$O ≈ -5.6 ‰) than the other, which we will call RefB ($\delta^2$H ≈ -69.8 ‰, $\delta^{18}$O ≈ -9.7 ‰), with the isotopic difference between the two reference waters being approximately 29.3 ‰ and 4.1 ‰ for $\delta^2$H and $\delta^{18}$O, respectively. To test whether smaller sample volumes were affected more substantially by vapor mixing and evaporation, we alternated the sample volumes between 200 and 400 ml. Thus, the carousel of each ISCO sampler contained three replicates of each possible





combination of the two reference waters (RefA vs. RefB), the two sample volumes (200 vs. 400 ml), and two bottle types

(open vs. retrofitted with evaporation protection).

We placed four additional sample bottles into the center of each autosampler carousel on day 1 of the experiment. Two of these bottles contained 200 ml of RefA water and the other two bottles contained 200 ml of RefB water; all four bottles were tightly sealed.


The bottom compartment of the autosampler (containing the sample bottles) and the middle compartment (containing the pump and control unit) remained attached for the entire duration of the experiment. Sample bottles were weighted at the start and end of the experiment to track potential changes in water volumes. After 21 days, the ISCO samplers were opened and all bottles were retrieved. We transferred 1.5 ml of the liquid sample water from each bottle into glass vials with screw

caps and stored them at 4 °C until isotope analysis.

**Experiment 3**

To assess the effectiveness of the retrofitted bottles under central European climatic conditions, we monitored evaporative fractionation in two ISCO autosamplers during 62 two-to-three-week sampling periods between October 2015 and

June 2019. For this purpose, we installed the ISCO autosamplers at two different locations in the northern Swiss pre-Alps: at the EIN site located near the city of Einsiedeln (8.75708°E, 47.13370°N, WGS84) at 910 m above sea level (m a.s.l) and at the ERL site located roughly 11 km southwest of Einsiedeln in the Erlenbach catchment (8.71502°E, 47.04249°N, WGS84) at 1228 m a.s.l.

At the beginning of each sampling period, we filled one tightly sealed, one open and one retrofitted sample bottle with 400 ml reference water each and placed them in the center of the ISCO carousel (the outer 24 bottles were reserved for conventional automatic precipitation sampling, not discussed here). The ISCO autosamplers remained at the field sites for roughly two to three weeks before all bottles were collected and replaced with new ones. After collecting the sample bottles, they were transported to the ETH Zurich laboratory and two 1.5 ml aliquots of sample water were transferred from each

bottle into glass vials with screw caps; the vials were stored at 4 °C until isotope analysis.

To identify potential drivers of evaporative fractionation effects during these sampling periods, we used on-site air temperature and relative humidity measurements. These measurements were provided by the Swiss Federal Office of Meteorology and Climatology (MeteoSwiss) for the EIN site and by the Swiss Federal Institute for Forest, Snow and

Landscape Research (WSL) for the ERL site. In addition, we used daily maximum, minimum and average values of air temperature and relative humidity to calculate the daily vapor pressure deficit (VPD = $e_s - e_a$) following Allen et al. (1998):



$$e_T^0 = 0.6108 \cdot exp(\frac{17.27 \cdot T}{T+237.3}) \ , \qquad (1)$$

$$e_s = \frac{e_{Tmax}^0 + e_{Tmin}^0}{2} \ , \qquad (2)$$

$\quad e_a = \frac{e_{Tmin}^0 \cdot H_{max} + e_{Tmax}^0 \cdot H_{min}}{2} \ , \qquad (3)$

where $e_T^0$ is the saturation vapor pressure at the air temperature $T$ (kPa), $e_s$ is the saturation vapor pressure (kPa), $e_a$ is the actual vapor pressure (kPa), $T$ is the air temperature (°C), $H$ is the relative humidity (-), and the indices *min* and *max* indicate the minimum and maximum values of temperature and relative humidity observed during any day. To compare these potential drivers with the isotope differences, we averaged the daily values of air temperature, humidity and VPD over the

individual sampling periods.

**2.3 Stable water isotope analysis and isotopic differences**

For Experiment 3, all water samples collected between 6 October 2015 and 13 December 2017 were analyzed at the laboratory of the Swiss Federal Institute for Forest, Snow and Landscape Research (WSL) with an Isotopic Water Analyzer LGR IWA-45-EP (Los Gatos Research, ABB Los Gatos Research, San Jose, California, USA) with a measurement precision

of 0.5 ‰ for $\delta^{18}O$ and 1 ‰ for $\delta^2H$. All of Experiment 3's samples collected after 13[th] December 2017, and all water samples of Experiments 1 and 2, were analyzed with a Cavity Ring-down Spectrometer at the ETH Zurich laboratory (L2140-*i* liquid isotope analyzer, Picarro Inc., Santa Clara, CA, USA) with a measurement precision of 0.2 ‰ for $\delta^{18}O$ and 1 ‰ for $\delta^2H$. All isotope values in this study are reported in $\delta$-notation relative to Vienna Standard Mean Ocean Water (V-SMOW) and the measurement uncertainty is provided as standard deviations calculated from 2 to3 repeated injections of

each sample.

To quantify the isotopic change in the water samples, we calculated the isotopic difference ($\Delta^{18}O$ and $\Delta^2H$, ‰) between the water sample at the end of the storage period, and the reference water at the beginning of the storage period:

$$\Delta \ ^iE = \delta \ ^iE_{sample} - \delta \ ^iE_{reference} \ , \qquad (4)$$

where $\delta \ ^iE_{sample}$ and $\delta \ ^iE_{reference}$ are the delta values of the isotope $\ ^iE$ in the sample water or the reference water, respectively. For Experiment 1, we compared the isotope composition of the water samples from the open and retrofitted bottles ($\delta^2H_{sample}$, $\delta^{18}O_{sample}$) to the isotope composition of the reference water from the storage tank ($\delta^2H_{reference}$, $\delta^{18}O_{reference}$). Because each second day we collected one reference water sample from the tank and filled one open and one retrofitted bottle with reference water, the comparison of the isotopic changes ($\Delta \ ^iE$) of samples from bottles with and

without evaporation protection assesses the effectiveness of the retrofitted sampler bottles in protecting against evaporative enrichment. For Experiment 2, we compared the isotopic composition of the RefA and RefB water samples from the various open and retrofitted bottles ($\delta^2H_{sample}$, $\delta^{18}O_{sample}$) to the isotopic composition of the closed sample bottles with the





corresponding reference water RefA or RefB ($\delta^2\text{H}_{\text{reference}}$, $\delta^{18}\text{O}_{\text{reference}}$). For Experiment 3, we compared the isotopic

composition of each sampling period's water samples from the open and retrofitted bottles ($\delta^2\text{H}_{\text{sample}}$, $\delta^{18}\text{O}_{\text{sample}}$) to the

isotope values of the reference water (i.e., $\delta^2\text{H}_{\text{reference}}$, $\delta^{18}\text{O}_{\text{reference}}$) for each sampling period.

## 3 Results

### 3.1 Laboratory evaluation of the evaporation protection method: Experiment 1

During Experiment 1, humidity outside the ISCO autosampler stayed relatively constant at approximately $11 \pm 3$ %

(mean±1standard deviation), while it continuously increased inside the lower ISCO compartment from 33 % to 100 %

between day 1 and 13, and then remained at 100 % until the end of the experiment (Figure 2a). Air temperature outside the

ISCO sampler was around $35 \pm 1$ °C with distinct diurnal variations (1.2 °C temperature drop at the beginning of the 4th day

was caused by moving the temperature logger to a more representative position inside the climate chamber). The air

temperature inside the ISCO housing was $36 \pm 1$ °C and did not exhibit strong diurnal patterns.


The sample bottles stored outside the autosampler (i.e. at 11 % relative humidity) experienced different degrees of

evaporative fractionation between the start and end of the monitoring period (Table 1): while evaporative fractionation was

insignificant in the closed bottle, we observed isotopic enrichment in both the retrofitted and the open bottle. Enrichment

was substantially stronger for the sample in the open bottle (a change of roughly 100 ‰ in $\delta^2\text{H}$ and 22 ‰ in $\delta^{18}\text{O}$ within

12 days), compared to the sample in the retrofitted bottle (a change of 9 ‰ in $\delta^2\text{H}$ and 2 ‰ in $\delta^{18}\text{O}$ over 24 days). The open

bottle had been sampled already on day 12 of the experiment because we had observed substantial evaporation by then. At

the end of the monitoring period (day 24), the water from the open control bottle had evaporated completely, while the loss

of water volume was small in the retrofitted control bottle (Figure 3).

The $\delta^2\text{H}$ values of the water samples from inside the ISCO sampler show that evaporative fractionation differed between

samples from open and retrofitted bottles, and also varied with storage duration (Figure 2b and c). The isotopic changes

(eq. 4) were generally smaller for samples in retrofitted bottles compared to those in the open bottles. For $\delta^2\text{H}$, the isotopic

changes ($\Delta^2\text{H}$) in the retrofitted bottles were mostly close to 0 ‰ independent of storage duration, while the isotopic changes

in the open bottles ranged up to 5 ‰ (Figure 2b). For $\delta^{18}\text{O}$, we obtained less clear fractionation signals, and while the

enrichment was always greater in $\delta^{18}\text{O}$ samples of open bottles compared to those in retrofitted bottles, we observed isotopic

depletion of up to 0.3 ‰ for samples filled early on in the experiment (data for $\delta^{18}\text{O}$ are presented in Figure S1).





Water samples filled before day 14 of the experiment, i.e. samples with more than 10 days of storage time, showed
substantially larger $\Delta^2$H values in the open bottles than in the retrofitted bottles; i.e. samples in open bottles experienced
stronger enrichment (Figure 2b). Conversely, samples introduced after day 16, and thus stored for 8 days or less,
experienced little or no evaporative fractionation, independent of the bottle type (i.e., $\Delta^2$H was not significantly different
from zero). This decrease in evaporative fractionation in the later samples may have been due to the increase in relative
humidity inside the ISCO compartment during the experiment, since the relative humidity inside the ISCO housing reached a
value of approximately 90 % on day 10 and 100 % on day 12. Surprisingly, samples filled on days 10 and 12 showed
stronger enrichment than those filled on adjacent days, both for open and retrofitted bottles. Because this isotope effect
occurred in both bottle types , it cannot be attributed to a specific process; it may have been related to spillage during sample
handling or interferences in the isotope analyser.

Figure 2c compares the $\Delta^2$H values of samples from open and retrofitted bottles against the average relative humidity inside
the ISCO sampler. While the change in relative humidity inside the ISCO did not seem to affect the samples in the retrofitted
bottles, we obtained a nearly linear relationship for the open bottles indicating a 2.8‰ enrichment per 10% decrease in
relative humidity ($p < 0.005$). Such a relationship is expected, because the vapor phase in the open bottle is in exchange with
the vapor phase inside the ISCO housing, and evaporation from the liquid phase is generally faster when water vapor
concentrations in the gas phase are lower (assuming constant temperature). Due to the much smaller contact area between
the liquid and vapor phases in the retrofitted ISCO bottles, vapor exchange was reduced and evaporation from the liquid
sample was much smaller (even when relative humilities were below 90 % inside the ISCO housing; Figure 2c). Retrofitting
made little difference at ≈100 % humidity, when vapor-pressure deficits, and thus evaporation rates, were minimal. Relative
humidity outside the ISCO and temperatures inside and outside the ISCO remained nearly constant throughout the
monitoring period, so no relationship with the observed isotopic composition could be identified and their effect on
evaporative fractionation in this laboratory experiment could not be assessed.

In spite of the high temperatures and low relative humidity, the observed fractionation effects during Experiment 1 were not
excessively large. This may have been due to some limitations in the setup. For one, we did not use the built-in peristaltic
pump of the ISCO system to fill our samples, which follows the protocol for collecting precipitation samples but is not
suitable for streamwater grab sampling. In case of streamwater sampling, the pump is used and the tubing between sampling
location and pump is flushed with air before sampling. This process likely results in an intake of air into the ISCO and
consequently enhanced vapor exchange with the surrounding atmosphere, which may enhance isotopic fractionation of the
collected water sample. In addition, we could not measure the water volumes in the sample bottles at the time of filling and
at the end of the experiment because we did not want to open the ISCO sampler during the experiment period. While we
took care to fill exactly 400 ml reference water into each sample bottle through the distributor arm, we cannot exclude that





some spillage occurred inside the ISCO during the filling procedure. It is therefore not possible to assess the exact amount of sample volume that was lost due to evaporation.

## 3.2 Assessing the effect of evaporative fractionation and mixing: Experiment 2

Ambient conditions during Experiment 2 were colder and more humid compared to Experiment 1, and substantially more variable. Outside the ISCO autosamplers, air temperature (mean ± standard deviation) was 13.3±6.2 °C in the outdoor setting and 18.6±4.7 °C indoors, while relative humidity was 73.5±23.0% outdoors with distinct daily fluctuations and 44.8±8.9% indoors. Temperature and relative humidity measured inside the autosampler housings exhibited similar but damped diurnal patterns (see Figure 4a, b). The temperature and relative humidity inside the outdoor ISCO were

16.7±6.7 °C and 86.0±13.6%, respectively. For the indoor ISCO, the respective values were 18.2±2.9 °C and 96.7±2.7%. In contrast to Experiment 1, the relative humidity inside the ISCOs did not increase gradually to 100 % over several days but remained high throughout the experiment, probably because all sample bottles were filled from the start, instead of successively as in Experiment 1.

The changes in isotopic composition, $\Delta^2H_{RefA}$ and $\Delta^2H_{RefB}$, were calculated following eq. (4) with *sample* being RefA or RefB water in the open or retrofitted bottles and *reference* being RefA or RefB water in the closed bottles. We observed no significant change in the isotopic composition of samples in retrofitted bottles (both $\Delta^2H_{RefA}$ and $\Delta^2H_{RefB} \approx 0$ ‰, red filled markers in Figure 4c-f; results for $\delta^{18}O$ were similar, see Figure S2), whereas the isotopic composition of samples in open bottles changed over the course of the experiment by up to 10 ‰ (blue open markers in Figure 4c-f). The observed change

in isotopic composition in open bottles was more pronounced for smaller sample volumes (comparison of circles and diamonds in Figure 4c-f). It was also larger in the outdoor setting compared to indoor conditions (comparison of Figure 4c, e with Figure 4d, f), even though the average temperature was lower in the outdoor setting. This may indicate that the average temperature is less important for causing isotope effects than the magnitude of the temperature fluctuations, or it may reflect the greater potential for wind-driven ventilation of the outdoor ISCO. Samples which underwent a stronger

change in isotopic composition also experienced a greater loss of sample volume between the start and end of the experiment. We observed a larger decrease in sample volumes in open bottles compared to the retrofitted sample bottles, a larger decrease in sample volumes in the outdoor setting compared to the indoor setting, and a larger relative decrease in the 200 ml samples compared to the 400 ml samples (see Figure S3 in the Supplement).

In the outdoor setting, the samples in the open bottles became isotopically heavier, with larger changes observed in the 200 ml samples than in the 400 ml samples (e.g., $\delta^2H$ in RefB water increased by 9.6 ‰ and 3.9 ‰ in 200 ml and 400 ml samples, respectively; Figure 4e), likely due to evaporative fractionation. For the open bottles in the indoor setting,





however, RefA samples became isotopically lighter by about 1 ‰ in $\delta^2$H (-1.2 ‰ and -0.9 ‰ for 200 ml and 400 ml, respectively; Figure 4e) while RefB samples became roughly 2 ‰ heavier (2.6 ‰ and 1.6 ‰ for 200 ml and 400 ml,

respectively; Figure 4f). The isotopic lightening of RefA samples may be explained by mixing in the vapor phase of isotopically heavier RefA water with isotopically lighter RefB water, and subsequent condensation in both samples. This isotopic exchange should make RefA isotopically lighter and RefB isotopically heavier, in addition to any isotopic fractionation due to net evaporative losses from both samples. Thus a large part of the observed enrichment in RefB water in the indoor setting may have been due to isotopic exchange with the heavier RefB water, in addition to any evaporative

fractionation. A likely reason why the mixing effect was only visible in the indoor setting may be that evaporation was smaller compared with the outdoor setting; the greater evaporative losses (and thus evaporative fractionation) in the outdoor setting may have overprinted the vapor mixing effect. In either case, mixing and/or evaporative fractionation only affected the isotopic composition in the open sample bottles, while no measurable effect was observed in samples from the retrofitted bottles.


We can quantify the isotopic changes due to mixing and evaporative fractionation in the different settings under the assumption that a) evaporative fractionation and mixing have additive effects, b) the per mil change due to evaporative fractionation is the same for RefA and RefB, and c) mixing has an exactly inverse effect on the two waters (i.e., it results in the same degree of isotopic depletion in the heavier RefA and enrichment in the lighter RefB):

$$m_{RefA} + f = \Delta^2 H_{RefA} \quad , \qquad (5)$$
$$m_{RefB} + f = \Delta^2 H_{RefB} \quad , \qquad (6)$$
$$m_{RefA} = -m_{RefB} \quad , \qquad (7)$$

where mixing-induced isotopic change is denoted by $m$ for RefA and RefB waters, and the change in isotopic composition due to evaporative fractionation is denoted by $f$.


The results of this analysis are illustrated in Figure 5 for $\delta^2$H, confirming that isotope effects due to mixing and evaporation are small in samples from the retrofitted bottles (red filled markers). In the open bottles (blue open markers), the isotope effects due to evaporative fractionation were between 1.5 to 2 times larger than the isotopic change due to mixing in the outdoor setting, but fractionation was less important than mixing in the indoor setting. Both the mixing- and fractionation-

induced isotope effects were roughly twice as large in the 200 ml sample volumes compared to the 400 ml sample volumes (diamonds vs. circles, respectively). Applying eqs. (5)-(7) to $\delta^{18}$O yielded similar results (see Figure S4 in the Supplement).

In summary, the results of Experiment 2 confirm the findings from Experiment 1 that the retrofitted ISCO bottles efficiently protect the collected water samples from undergoing isotopic changes due to both evaporative fractionation and vapor

mixing.





### 3.3 Evaluation of the evaporation protection in the field

During the field experiment (Experiment 3, October 2015 to June 2019), we observed distinct seasonality in air temperature, but no seasonal pattern in relative humidity at both field sites, and slightly higher temperatures and humidity at the EIN site (Figure 6). At both field sites, the vapor pressure deficit (VPD), which is strongly correlated with air temperature, peaked
around June and July and was lowest in December and January. Because these climatic variables exhibited very similar behavior at both field sites, we decided to pool the isotope data sets from both sites for analysis.

The isotope data from 8 March 2016 were excluded from this analysis because the water samples in the retrofitted bottles were isotopically lighter than the reference water (e.g., $\Delta^2H$ ranged from -4.7 to -3.5 ‰). In addition, we removed the data points from 14 February 2017 from our analysis because of an anomalous $\delta^{18}O$ measurement of the water sample from the
retrofitted bottle at the ERL site.

The field experiment with the open and retrofitted bottles in two ISCO samplers resulted in 244 usable samples (i.e., 61 samples per site and bottle type) for which the storage duration varied between 12 and 23 days. The isotopic differences of the samples in open bottles relative to the reference water (mean±1 standard error) were 1.45±0.22‰ for $\Delta^2H$ and
0.27±0.05 ‰ for $\Delta^{18}O$, and thus deviated significantly from zero. Conversely, when the retrofitted bottles were used, the isotopic differences were statistically not significantly larger than zero, i.e., 0.10±0.11 ‰ for $\Delta^2H$ and 0.05±0.03 ‰ for $\Delta^{18}O$. Figure 7 and Figure S5 show that the isotopic difference of the samples in open bottles relative to the reference water was positively correlated with average air temperature, and thus with VPD. Pearson correlation coefficients between air temperature (average, minimum and maximum) and $\Delta^2H$ were $r>0.70$ ($p<0.001$), and $r>0.60$ ($p<0.001$) for the same
relationships with $\Delta^{18}O$. For VPD, the correlation coefficients were $r>0.56$ ($p<0.001$) for both isotopes. No statistically significant relationships were evident for the samples from the retrofitted bottles ($r<0.17$, $p>0.1$; see also the red data points in Figure 7 and S5). Similarly, the maximum change in air temperate and relative humidity (as an indicator for climatic variability at our field sites) were positively correlated with the isotopic difference only for the open bottles (the relationship was statistically significant only for the change in air temperature, with $r>0.40$ and $p<0.0001$; Figure 8 and Figure S6).

Overall, our results indicate that the retrofitted sample bottles significantly reduced isotopic fractionation compared to the open sample bottles when deployed over two- to three-week periods under the ambient climatic conditions at our two field sites.



## 4 Practical implications

In the three experiments presented above, we assessed how storage duration, temperature and humidity fluctuations, and sample volume influenced isotopic shifts due to evaporative fractionation and vapor mixing in samples stored inside the ISCO autosampler. In all three experiments we found that the observed change in isotopic composition was substantially smaller in sample bottles retrofitted for evaporation protection.

We can use the relationship between isotopic fractionation and air temperature from Experiment 3 to estimate the expected isotopic change in the laboratory during Experiment 1. If we apply the linear regression slopes shown in Figure 7b to estimate the isotopic change in the open bottle at the average air temperature of 35 °C maintained during Experiment 1, we obtain an expected change in isotopic composition of $\Delta\,^2H=9.4\pm1.1$ ‰ ($\pm$ 1 standard error) and $\Delta^{18}O=1.6\pm0.3$ ‰. These values are substantially larger than the differences obtained after 24 days of Experiment 1 (i.e., 5 ‰ in $\Delta^2H$ and 1 ‰ in $\Delta^{18}O$; Sect. 3.1). The larger isotopic change observed during the field deployment of Experiment 3 may be attributed to more variable climatic conditions (e.g., due to diurnal temperate variations), and possibly also to ventilation by wind, whereas the sampler was placed in a windless chamber with constant temperature and relative humidity during Experiment 1. This hypothesis is supported by the results from Experiment 2; here, the average temperature was lower in both the indoor and outdoor settings compared to Experiment 1, but the change in isotopic composition was comparable (in Experiment 1 we observed an increase in $\delta^2H$ of approx. 3 ‰ in the open sample bottle with a storage duration of 21 days; in Experiment 2, the increase was 1.7 ‰ and 4 ‰ in the indoor and outdoor settings, respectively, for the 400 ml samples of RefB). Larger temperature and humidity contrasts due to diurnal fluctuations in outdoor conditions may have resulted in repeated evaporation and condensation inside the ISCO housing, and in enhanced vapor exchange between the sample bottles and the outside atmosphere, in contrast to the very stable conditions during Experiment 1.

Our evaporation protection reduced the contact area between the water surface in the sample bottle and the atmosphere inside the ISCO autosampler by a factor of approximately 5500 (comparing the cross-sectional area of the bottle to that of the silicone tube attached to the syringe), and also the area for diffusion of vapor through the bottle opening by a factor of approximately 1300 (comparing the area of the bottle opening to the cross-sectional area of the silicone tube). Consequently, isotopic fractionation and mixing should be substantially reduced in samples in retrofitted bottles compared to those in open sample bottle. However, because the syringe housing does not entirely seal the ISCO sample bottle (because air needs to be released when water samples are introduced into the bottle), some vapor exchange may still occur between the sample bottle and the inside of the ISCO compartment. This vapor exchange will likely be stronger if air temperature is high and relative humidity inside the ISCO housing is low (Experiment 1). Experiment 2 also suggested that strong diurnal variations or windy conditions will also increase vapor exchange and consequently evaporative fractionation. In central European climates, such conditions may occur during extremely dry and warm summer days so that automatically collected water





samples should be retrieved sooner than 24 days if possible. However, Experiment 1 showed that in the absence of wind, the relative humidity inside the ISCO can build up over time, even if the relative humidity outside is very low.


We furthermore showed that in open bottles, 400 ml samples exhibited smaller isotope effects than 200 ml samples, simply because the ratio between the water volume affected by mixing and fractionation (i.e., the uppermost water layer that is in exchange with the atmosphere) and the total sample volume is two times smaller for the 400 ml sample than for the 200 ml sample (Experiment 2). We therefore recommend that streamwater samples collected with our evaporation protection

method should comprise at least 400 ml (but note that due to the narrow tube, care has to be taken to not exceed a filling rate of approximately 100 ml min$^{-1}$). When collecting precipitation samples, larger sample volumes can be achieved by using larger funnels; e.g., 1 mm of rain collected with a 45-cm diameter funnel results in approximately 160 ml sample volume, while a 20-cm diameter funnel would only yield around 30 ml.

While we have discussed the performance of the retrofitted ISCO 1-litre sample bottles with respect to stable water isotopes, the new bottle design may also be useful for water quality studies. Experiment 2 showed that evaporation from open sample bottles results in reduced water sample volumes, implying evapo-concentration of solutes in the samples. The importance of this effect likely depends on the storage duration and the sample volume, and will therefore be greater for small samples and for samples collected early on in the sampling period. As a result, water quality data from water samples automatically

collected over periods of days and weeks may not be directly comparable. The results from Experiment 2 suggest that our retrofitted sample bottle may reduce evapo-concentration effects in the water samples. The presented evaporation protection modification could also be combined with a gravitational filtration system similar to the ones presented by Kim et al. (2012) added between the syringe outlet and the silicone tubing, but further studies would be needed to assess this in a more systematic manner.

**5 Conclusions**

We tested whether retrofitting the sample bottles of the conventional ISCO 6712 Full-size Portable Sampler (Teledyne ISCO., Lincoln, US) with a modified syringe housing and silicone tube reduces evaporative fractionation and vapor mixing in water samples collected for subsequent analysis of stable water isotopes ($\delta^2$H, $\delta^{18}$O). Laboratory and field tests under different temperature and humidity conditions showed that water samples in bottles with evaporation protection were far less

altered by evaporative fractionation and vapor mixing than samples in conventional open bottles.

We showed that retrofitting sample bottles with a modified syringe housing and silicone tube is a cost-efficient approach to upgrade autosampler bottles to protect water samples from isotopic fractionation during storage in the field. The setup



described here can likely be adapted without difficulty (e.g., by using a different syringe size) to be compatible with bottles

in other autosamplers, such as the Maxx P6L-Vacuum System (Maxx GmbH, Rangendingen, Germany), or the smaller ISCO 6712C and 3700C Compact Portable Samplers (Teledyne ISCO., Lincoln, US) that use 500 ml sample bottles. These will require further testing, because the observed results partly depend on the size of the air space inside the automatic sampler compartment, which influences the buildup of humidity inside the autosampler. Different systems may also be more or less tightly sealed from the atmosphere, likely resulting in stronger or weaker vapor exchange.


ISCO autosamplers are generally available in many laboratories, but researchers may be reluctant to use them for isotope studies due to a fear of evaporative fractionation occurring in the water samples, particularly if sample volumes are small and/or weather conditions are dry and warm. Our inexpensive and robust method can be used to substantially reduce evaporative fractionation in water samples collected at daily or sub-daily frequency over periods of up to 24 days.

**Data availability**

The stable water isotope measurements from the three experiments are provided as supplemental material.

**Acknowledgements**

We wish to thank Nikos Anestis, Stephan Biber, Stefan Boss, Linus Ender, Özden Erden, Joël Frey, Selina Ilchmann, Vincent Marmier Daniel Meyer, Dominic Schori and Kari Steiner for their help in the field and in the laboratory, and
Alessandro Schlumpf and Massimiliano Zappa for their help with the development of the evaporation protection modification. JLAK acknowledges support from an ETH Zurich Postdoctoral Fellowship. JF was supported by the Swiss National Science Foundation SNF (grant PR00P2_185931).

**Author contributions**

JF, JLAK and JWK conceptualized the study. JF, JLAK, AR and BS performed the experiments and analyzed the isotope
data, JF, JLAK, and AR analyzed the datasets, JF and JLAK prepared the manuscript with contributions from all co-authors.

**Competing interests**

The authors declare that they have no conflict of interest.





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





**Tables**

**Table 1: Isotopic changes in the water samples relative to the references water in three different bottles that were stored outside the ISCO sampler during Experiment 1. Isotopic changes are expressed as mean±1 standard deviation.**

| Bottle type | Storage duration [days] | $\Delta^2$H [‰] | $\Delta^{18}$O [‰] |
|---|---|---|---|
| Closed | 24 | -0.42 ± 0.27 | -0.08 ± 0.12 |
| Retrofitted | 24 | 9.05 ± 0.54 | 1.63 ± 0.15 |
| Open | 12 [a] | 100.46 ± 1.04 | 21.97 ± 0.20 |

[a] For the open bottle, the change in isotopic composition between day 0 and day 12 is provided, because the water sample was fully evaporated from the open bottle by day 24.



**Figures**

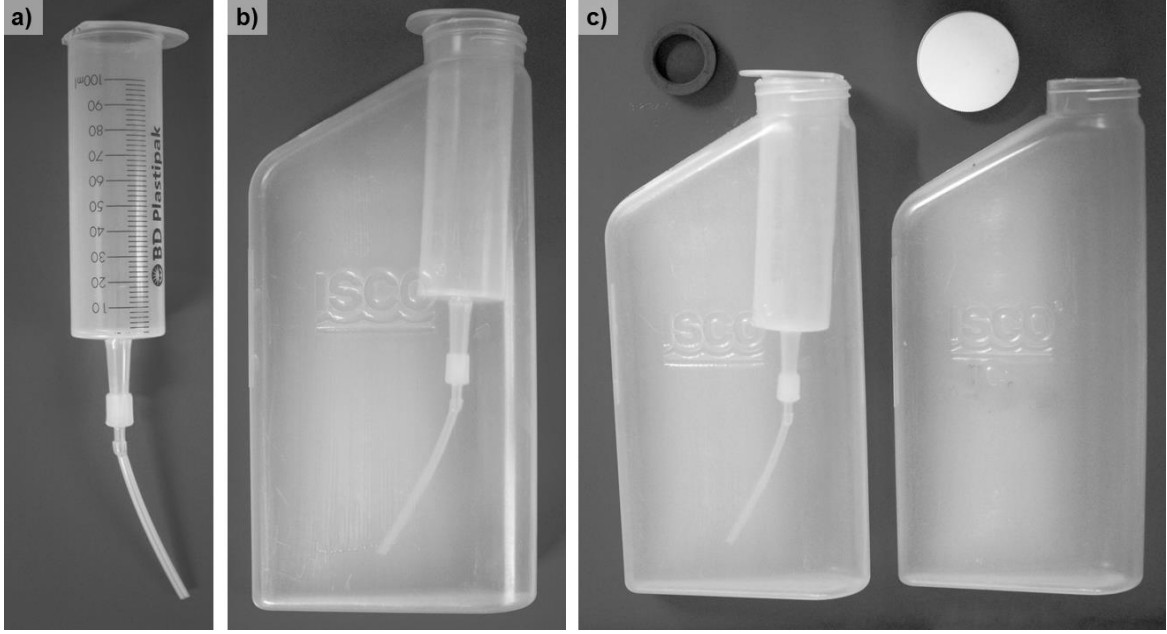

**Figure 1: (a) Modified 100 ml syringe housing with Luer-tip adapter and fitted silicone tubing for extending the syringe outlet to the bottom of the bottle; the barrel flange at the syringe housing was trimmed on the outer side to ensure the retrofitted bottles fit into the autosampler. (b) Retrofitted sample bottles with evaporation protection using the modified syringe shown in (a). (c) The**
**sample bottles with evaporation protection can be sealed for transport with black rubber piston stoppers (left); bottles without evaporation protection can be sealed with a screw lid (right).**

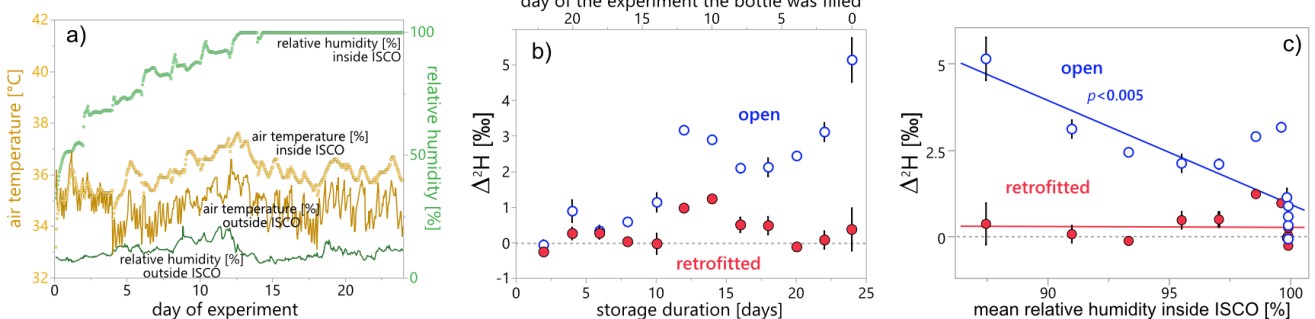

**Figure 2: (a) Evolution of air temperature (yellow) and relative humidity inside (light-green dots) and outside (dark-green line) the ISCO sampler during Experiment 1. Humidity outside the ISCO autosampler stayed relatively constant between 6% and 21%, whereas humidity inside the sampler increased over time to 100%. Average air temperature was similar inside and outside, but fluctuations were more pronounced outside of the autosampler. (b) The isotopic enrichment of the sample water relative to the reference water (here expressed as isotopic change $\Delta^2H$) was stronger for samples in open bottles, and increased with longer**
**storage durations. The isotopic change was calculated for each sample relative to the isotopic composition of the reference water in the storage tank on the day the bottle was filled (eq. 4). (c) Isotopic change in water samples relative to the reference water as a function of the mean relative humidity during the storage period of each sample. Linear regression (solid line) for the open bottles is statistically significant at $p<0.005$; the regression slope for the retrofitted bottles is not statistically different from zero. In panels (b) and (c), water samples in open bottles are marked with blue open circles, whereas water samples in retrofitted bottles are**
**marked with red filled circles. Error bars indicate the measurement uncertainty as ±1 standard deviation.**


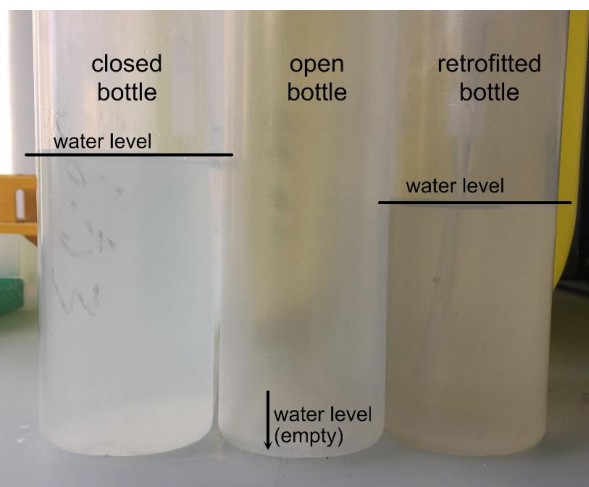

**Figure 3: Water levels in the control bottles after the 24-day laboratory experiment (Experiment 1). The water level in the closed bottle is identical to the water level in all three bottles at the start of the experiment. By the end of the experiment the water level in the retrofitted bottle had only decreased slightly, while the water from the open bottle was completely evaporated after approximately 12 days.**




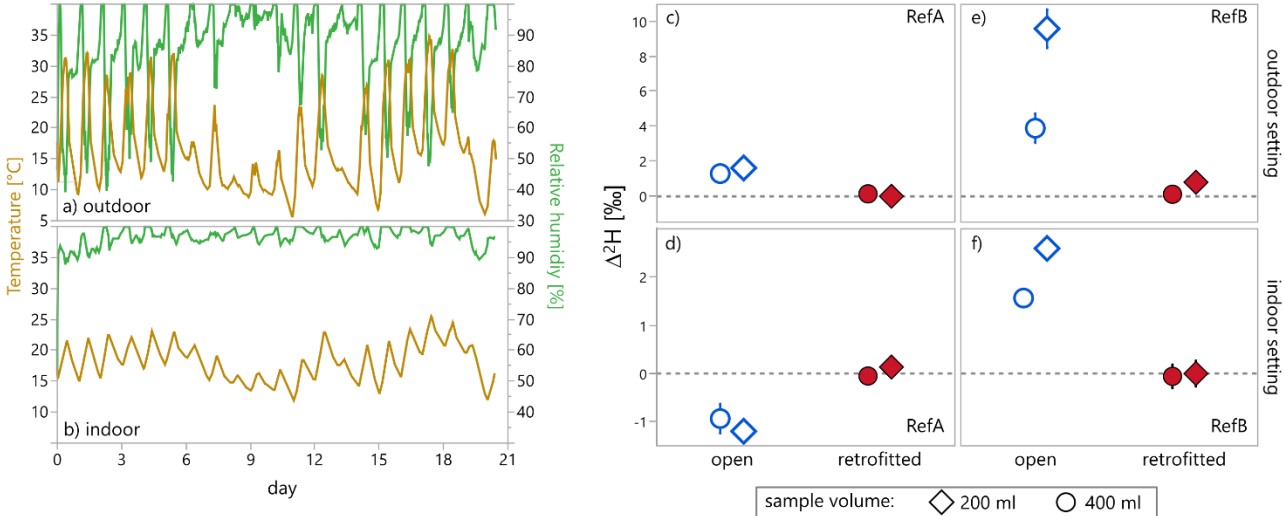

**Figure 4: (a,b): Temperature (yellow) and relative humidity (green) measured inside the ISCO samplers that were located oudoors (a) and indoors (b) over the 21 days of Experiment 2. (c-f): Mean change in isotopic composition of samples relative to the reference waters. Each data point is calculated from the three replicates of each combination of the two reference waters (RefA vs. RefB), the two sample volumes (200 vs. 400 ml), and two bottle types (open vs. retrofitted with evaporation protection). Please note the different y-axis scales between panels c), e) and d), f). Error bars denote standard errors of the three replicates of each**
**condition, and account for measurement uncertainty and the standard error of the sample means.**



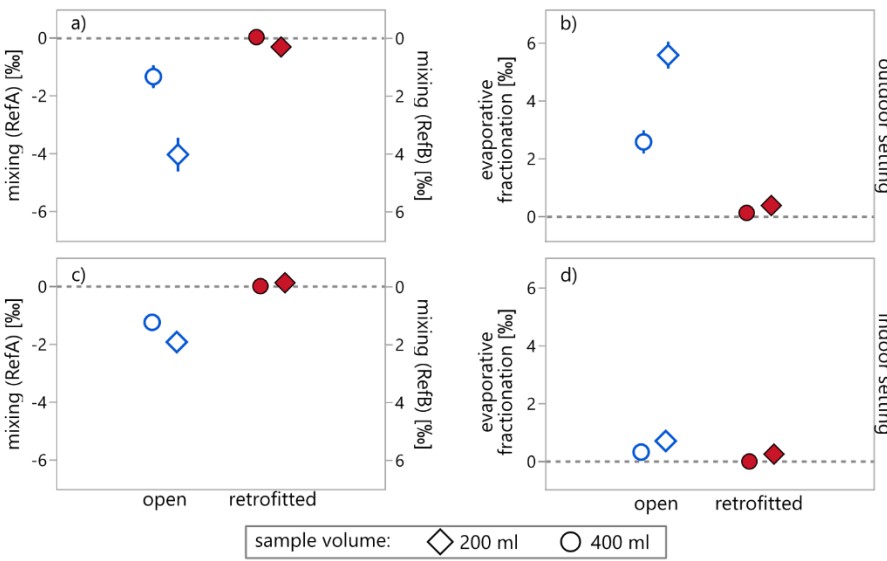

**Figure 5:** A comparison of the isotope effects due to mixing (panels a and c) and evaporative fractionation (panels b and d) in water samples stored outdoors (panels a and b) and indoors (c and d). Both mixing and evaporative fractionation effects are small in samples from the retrofitted bottles (filled red markers), and larger in samples from the open bottles (open blue markers). In addition, the isotope effects were larger for the 200 ml samples (diamonds) than for the 400 ml samples (circles). Error bars indicate standard errors determined from the three replicates of each combination of the two reference waters (RefA vs. RefB), the two sample volumes (200 vs. 400 ml), and two bottle types (open vs. retrofitted with evaporation protection), and they account for measurement uncertainty and the standard error of the sample means.






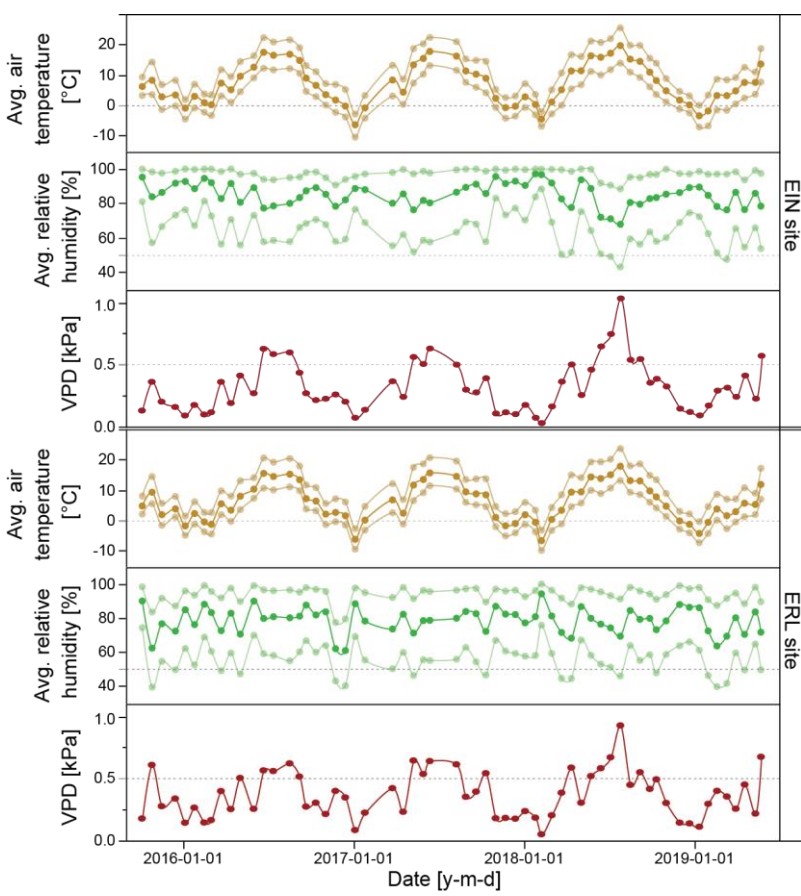

**Figure 6: Mean, minimum, and maximum values of air temperature and relative humidity, as well as vapor pressure deficit (VPD), averaged over two- to three-week storage periods during Experiment 3 at the two field sites EIN and ERL. Dashed horizontal lines in each panel indicate 0 °C air temperature, 50 % relative humidity and 0.5 kPa VPD for easier comparison between sites.**



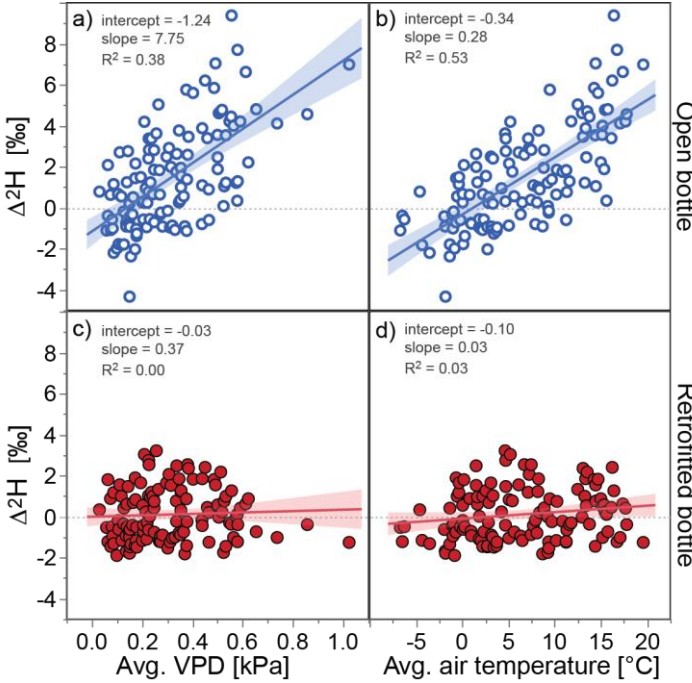

**Figure 7: Differences in δ²H (Δ²H) between samples in the ISCO bottles and their accompanying completely closed reference water samples over storage periods of 12-23 days during Experiment 3 at the EIN and ERL sites, and their relationships with the average vapor pressure deficits (VPD) and the average air temperatures during the respective storage periods. Open bottles show a substantial isotopic enrichment with higher VPD and air temperature, whereas retrofitted bottles do not indicate a systematic fractionation effect. No relationship with relative humidity was found. The uncertainties of the individual Δ²H values were on average 0.52 ‰; linear regression fits are indicated by solid lines and slope, intercept and R² values; the shaded areas represent the 95 % confidence intervals of the fitted line.**






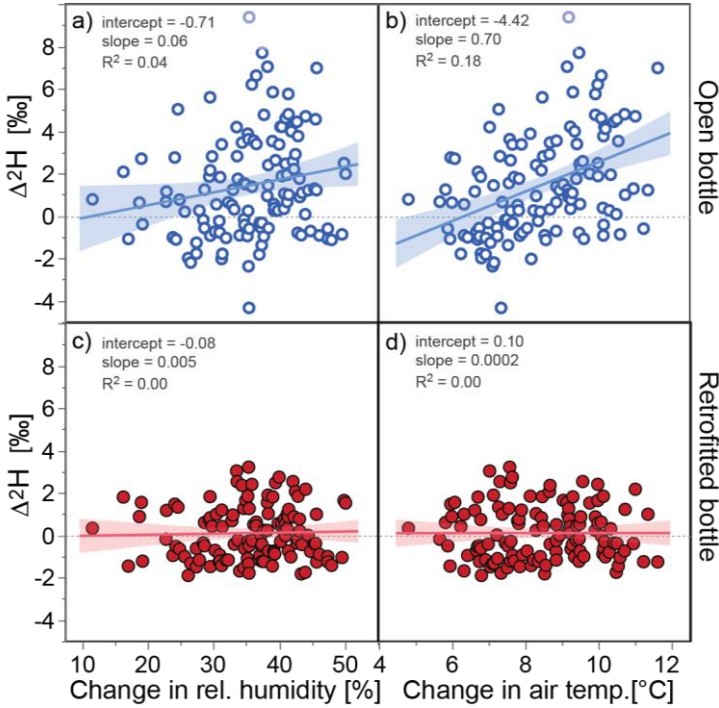

**Figure 8: Differences in δ²H (Δ²H) between samples in the ISCO bottles and their accompanying completely closed reference water samples over storage periods of 12-23 days during Experiment 3 at the EIN and ERL sites, and their relationships with the maximum changes in relative humidity and air temperature within the respective storage periods. Large changes in relative humidity resulted in some isotopic enrichment in the open bottles (open blue circles), but not in the retrofitted bottles (filled red circles). Open bottles showed the strongest isotopic enrichment when temperature contrasts were large (>10°C), whereas**
**retrofitted bottles seemed to be unaffected by temperature changes. The relationship between change in air temperature and Δ²H in open bottles was statistically significant ($r=0.42$, $p<0.0001$). The uncertainties of the individual Δ²H values were on average 0.52 ‰; linear regression fits are indicated by solid lines and slope, intercept and R² values; the shaded areas represent the 95 % confidence intervals of the fitted line.**