# Peer review of "Technical Note: Evaluation of a low-cost evaporation protection method for portable water samplers"

_Hydrology and Earth System Sciences, 2020_

## Referee Comment (RC1) · Nils Michelsen (Referee) · 18 Aug 2020

Dear colleagues,

With great interest, I have read your manuscript on the design of an evaporation reduction method for automatic water samplers, facilitating their use in isotope hydrology studies. Given the popularity of these samplers, the topic is, in my opinion, relevant for the HESS community.

The simple concept is described in detail and since the required parts are low-cost and readily available, the method is easy to apply (if the reader has access to an automatic

sampler).

The experimental designs are well-thought-out and the corresponding results indicate that the suggested mechanism is indeed capable of reducing post-sampling evaporation, although there are limitations (certain climatic conditions, storage times).

The manuscript is also well structured and written.

I do have a few remarks (see below), but overall I think the manuscript should be published as a Technical Note in HESS, after minor revisions.

Introduction: I think Williams et al. (2018) should be cited somewhere in the introduction. They have tackled the topic of post-sampling evaporation from automatic samplers before and should be given credit for their efforts. As their approach was rather different from yours (other evaporation reduction methods; experiments in insulated boxes instead of real automatic samplers), mentioning their work will provide additional context (and justification) for your work.

Line 47: That the tube dips into the collected water is obviously the most important aspect of this mechanism, but for sake of completeness, you could also mention the complementing aspect of the Gröning et al. (2012) design, i.e. the pressure equilibration tube (e.g. "Tube-dip-in-water collector with pressure equilibration", IAEA 2014). To additionally cite the pioneer work by IAEA (2002) is a good idea, but the reference is missing in the reference list.

Lines 61-63: This sentence is a bit misleading. It suggests that only changes in air temperature and humidity would cause problems, but evaporation and vapor mixing would also occur if temperature and humidity remained constant.

Line 78: I suggest to refer to Figure 1a directly after "9 cm in length"

Line 80: The word "tightly" is misleading here. If the syringe housing is really plugged tightly into the bottle opening (i.e. air-tight), the whole mechanism would not work anymore. When additional water is supposed to flow into the bottle, air needs to be

displaced and has to leave the bottle. You do mention this aspect later (Section 4, line 412-413), but this should be made clear here. You could even explicitly mention that you skip the "pressure equilibration tube" of the original concept (Gröning et al., 2012; IAEA 2002, 2014; see comment above).

Lines 85-91: I appreciate that this potential pitfall is highlighted and that you provide a maximum filling rate. I can imagine that debris (e.g. sediment, insects) may further reduce the tubing diameter (1 mm is quite small) in a field setting. Hence, it is probably a good idea to use a screen at the intake (e.g. a funnel screen) and maybe you want to mention this explicitly somewhere.

Line 104: For sake of consistency, this part should read "...during 62 two-to-three-week cycles..." (as in lines 179 and 382)

Line 111: Please provide some details on the climate-controlled chamber. If this was a commercial chamber, please mention model and manufacturer. If it was a custom-made chamber, please indicate this.

Line 113: Please also mention details on the temperature and humidity loggers (model, manufacturer, precision). Sometimes, it is these technical details that matter for the reader, particularly if they want to conduct similar experiments (as indirectly suggested in line 452). Knowing which logger can cope with such conditions (rel. humidity of up to 100 %), can be valuable.

Line 217: Wouldn't $\Delta\delta$18O and $\Delta\delta$2H be more accurate?

Line 230: Please specify what the reference water is here (tightly sealed bottles).

Lines 237-239: The 1°C temperature difference between the climate chamber and the ISCO (inside the climate chamber) is a bit surprising, particularly because the temperatures matched quite well before the logger as moved "to a more representative position".

Table 1: Probably, this should read "... relative to the reference water...".

Lines 264-267: The isotopic shifts observed for days 10 and 12 are indeed remarkable and I wonder if it is a coincidence that this was about the time when a humidity of about 100 % was reached. Could condensing water play a role here (i.e. liquid water drops flowing into the bottles)? Concerning the possible explanations that are given: I do not understand how spillage could explain the phenomenon.

Fig. 2: I am a bit confused about the x-axis in Fig. 2c. I was able to match all blue data points in 2b and 2c and conclude that some of the blue points in 2c represent the first days of the experiment (e.g. days 0 and 2). During this time, humidity was around 50 % according to 2a, but in 2c the x-axis only shows humidities above 87.5 %.

Line 282: Please make clear that the "low relative humidity" refers to the chamber conditions (not the conditions inside the ISCO).

Lines 296-298: If these temperatures and humidities are not shown anywhere, please indicate this to avoid confusion.

Lines 312-314: If wind plays a role strongly depends on the geometry of the setup. A possible scenario that is not mentioned here so far is temperature-triggered gas volume changes. In contrast to Experiment 1 (climate chamber), Experiment 2 was characterized by significant daily temperature fluctuations (20°C and more inside the ISCO; outdoor setting; Fig. 4a). Upon heating (daytime), the air inside the ISCO expanded and some of the (moist) air was pushed out of the device. When temperatures dropped (at night), the opposite happened, i.e. the air in the ISCO contracted and sucked in fresh air from outside. This "sampler breathing" probably happened on a daily basis, resulting in a greater air exchange, which in turn (apparently) caused lower humidities inside the ISCO (and more evaporation). Maybe this effect could have played a role here as well.

Lines 328-329: This sentence is confusing. Do you mean "…exchange with the heavier RefA water…"?

Lines 370-372: I am not sure if this statistical summary is sufficient. The mean value is close to zero, but I think you should at least mention that the isotopic shifts showed a fairly large scatter and ranged from about -2 (i.e. isotopic depletion) to about +3 ‰ (isotopic enrichment) in case of $\delta$2H (Fig. 7c and 7d). In case of $\delta$18O, the maximum deviations are remarkable as well. Here, $\Delta\delta$18O scatters between about 0.7 and about +1.2 ‰ (Fig. S5c and S5d). Both values are clearly beyond the analytical precision.

Practical implications: Here, the wind issue is stressed again. Maybe the "sampler breathing" (see above) also deserves to be mentioned here. If it really played a role, a practical consequence may be the need of a thermal insulation of the ISCO (reducing the temperature fluctuations inside the device).

Practical implications/Conclusions: The partly significant $\Delta\delta$2H and $\Delta\delta$18O values observed in Experiment 3 (see above) underscore the relevance of such "field controls". These pre-filled bottles (known isotopic composition and mass), placed into the automatic sampler upon field installation, allow for a hindsight evaluation of the samples' isotopic integrity. Although their advantage is somewhat obvious, it may be a good idea to explicitly recommend this technique to the reader.

I hope you will find these comments useful.

Best regards,

Nils Michelsen

References

Williams, M.R., Lartey, J.L., Sanders, L.L., 2018. Isotopic ($\delta$18O and $\delta$2H) Integrity of Water Samples Collected and Stored by Automatic Samplers. Agricultural & Environmental Letters, 3(1), 1-5. https://doi.org/10.2134/ael2018.02.0009

---

## Short Comment (SC1) · 29 Aug 2020

**Authors' response to the interactive comment by Nils Michelsen (Referee)**
**on "Technical Note: Evaluation of a low-cost evaporation protection method for portable water samplers"**
**by Jana von Freyberg et al.**

Dear colleagues,
With great interest, I have read your manuscript on the design of an evaporation reduction method for automatic water samplers, facilitating their use in isotope hydrology studies. Given the popularity of these samplers, the topic is, in my opinion, relevant for the HESS community. The simple concept is described in detail and since the required parts are low-cost and readily available, the method is easy to apply (if the reader has access to an automatic sampler). The experimental designs are well-thought-out and the corresponding results indicate that the suggested mechanism is indeed capable of reducing post-sampling evaporation, although there are limitations (certain climatic conditions, storage times).

The manuscript is also well structured and written.

I do have a few remarks (see below), but overall I think the manuscript should be published as a Technical Note in HESS, after minor revisions.

**We thank Nils Michelsen for this positive evaluation of your work and his comments, which were very helpful for improving our manuscript. We have addressed all comments in detail below (in blue bold font).**

Introduction: I think Williams et al. (2018) should be cited somewhere in the introduction. They have tackled the topic of post-sampling evaporation from automatic samplers before and should be given credit for their efforts. As their approach was rather different from yours (other evaporation reduction methods; experiments in insulated boxes instead of real automatic samplers), mentioning their work will provide additional context (and justification) for your work.
**Thank you. We will add this reference to the introduction.**

Line 47: That the tube dips into the collected water is obviously the most important aspect of this mechanism, but for sake of completeness, you could also mention the complementing aspect of the Gröning et al. (2012) design, i.e. the pressure equilibration tube (e.g. "Tube-dip-in-water collector with pressure equilibration", IAEA 2014). To additionally cite the pioneer work by IAEA (2002) is a good idea, but the reference is missing in the reference list.
**We will include IAEA (2002) in the reference list. We will also add a statement to the methods section:**
**"*Because there remains a small gap between the syringe housing and the inner rim of the sampler bottle opening (i.e., not air-tight), pressure differences due to water flowing into the bottle will equilibrate with the outside conditions. In this way, our system does not require an external tube for pressure equilibration such as the "tube-dip-in-water collector" proposed by Gröning et al. (2012).*"**

Lines 61-63: This sentence is a bit misleading. It suggests that only changes in air temperature and humidity would cause problems, but evaporation and vapor mixing would also occur if temperature and humidity remained constant.
**We will re-formulate the sentence: "However, because the sample bottles remain open during the sampling period, *vapor exchange may occur between the sample water and the atmosphere inside the sampler housing, which may alter* the isotopic compositions of the water samples in the bottles."**

Line 78: I suggest to refer to Figure 1a directly after "9 cm in length"
**We will add the reference to Figure 1a after the end of the suggested sentence and change the sentence to:**
**"On the Luer tip, we fit a 1-mm inner diameter silicone tube approximately 9 cm in length, to reach the bottom of the sample bottle (Figure 1 _a_, b)."**

Line 80: The word "tightly" is misleading here. If the syringe housing is really plugged tightly into the bottle opening (i.e. air-tight), the whole mechanism would not work anymore. When additional water is supposed to flow into the bottle, air needs to be displaced and has to leave the bottle. You do mention this aspect later (Section 4, line 412-413), but this should be made clear here. You could even explicitly mention that you skip the "pressure equilibration tube" of the original concept (Gröning et al., 2012; IAEA 2002, 2014; see comment above).

**This is a good point. We will remove the term "*tightly*" from the sentence and clarify the setup by adding the explanation proposed above: "*Because there remains a small gap between the syringe housing and the inner rim of the sampler bottle opening (i.e., not air-tight), pressure differences due to water flowing into the bottle will equilibrate with the outside conditions. In this way, our system does not require an external tube for pressure equilibration such as the "tube-dip-in-water collector" proposed by Gröning et al. (2012).*"**

Lines 85-91: I appreciate that this potential pitfall is highlighted and that you provide a maximum filling rate. I can imagine that debris (e.g. sediment, insects) may further reduce the tubing diameter (1 mm is quite small) in a field setting. Hence, it is probably a good idea to use a screen at the intake (e.g. a funnel screen) and maybe you want to mention this explicitly somewhere.
**Good point. We will include a sentence after Line 86: "*In order to protect the evaporation protection from clogging with debris (e.g. sediment, insects, leaves) the streamwater intake or precipitation funnel can additionally be equipped with a screen.*"**

Line 104: For sake of consistency, this part should read ". . .during 62 two-to-three week cycles. . ." (as in lines 179 and 382)
**We will change this as suggested.**

Line 111: Please provide some details on the climate-controlled chamber. If this was a commercial chamber, please mention model and manufacturer. If it was a custom-made chamber, please indicate this.
**We will clarify this: "The ISCO autosampler was placed *on a heater inside a ventilated chamber* where the conditions were kept at approximately 35 °C air temperature and 11 % relative humidity." We will also change the term "climate-controlled chamber" to "*ventilated chamber*" throughout the text (four instances in total).**

Line 113: Please also mention details on the temperature and humidity loggers (model, manufacturer, precision). Sometimes, it is these technical details that matter for the reader, particularly if they want to conduct similar experiments (as indirectly suggested in line 452). Knowing which logger can cope with such conditions (rel. humidity of up to 100 %), can be valuable.
**The specifications of the temperature and humidity loggers are: RHT30 humidity-temperature data logger; EXTECH Instruments, FLIR Commercial Systems Inc., Nashua, NH, USA; measurement accuracy ±1 % rel. humidity and 0.5 °C temperature). We will include this information in the methods section of the revised manuscript.**

Line 217: Wouldn't $\Delta\delta18O$ and $\Delta\delta2H$ be more accurate?
**We decided to avoid double-delta notations for the sake of easier readability.**

Line 230: Please specify what the reference water is here (tightly sealed bottles).
**We will add this information to the sentence to be more specific: "…of the reference water (i.e., $\delta^2H_{reference}$, $\delta^{18}O_{reference}$ *of the water in the tightly sealed bottles*) for each sampling period."**

Lines 237-239: The 1°C temperature difference between the climate chamber and the ISCO (inside the climate chamber) is a bit surprising, particularly because the temperatures matched quite well before the logger as moved "to a more representative position".
**We will add a better explanation: "Air temperature outside the ISCO sampler was around 35 ± 1 °C with distinct diurnal variations (*a* 1.2 °C temperature drop at the beginning of the 4th day was caused by moving the *humidity*-temperature *logger from a position close to the heater* to a *higher* position *near the sampler's control unit to better represent the conditions* inside the *ventilated* chamber).**

Table 1: Probably, this should read ". . . relative to the reference water. . .".
**Correct, we will change this accordingly.**

Lines 264-267: The isotopic shifts observed for days 10 and 12 are indeed remarkable and I wonder if it is a coincidence that this was about the time when a humidity of about 100 % was reached. Could condensing water play a role here (i.e. liquid water drops flowing into the bottles)? Concerning the possible explanations that are given: I do not understand how spillage could explain the phenomenon.

**To check whether condensation might have played a role here, we calculated the dew point temperatures based on the Magnus formula and the parameter set from Sonntag (1990).  As can be seen in the figure below, the dew point was reached inside the ISCO sampler after day 12 of the experiment.  Even though this coincides with the day 100% relative humidity was reached, this would suggest that all samples filled on and after day 12 would also have been affected by condensation, which was not the case.  In addition, because of our sampling design, all previous samples (days 0-10) were exposed to the dew point conditions as well but did not show such large isotopic differences as those samples from days 10 and 12.**
**We therefore can only speculate about the larger isotope differences in the samples of days 10 and 12.  We removed the possible explanation of spilling and kept measurement errors during isotope analysis.**

[Figure]

Fig. 2: I am a bit confused about the x-axis in Fig. 2c. I was able to match all blue data points in 2b and 2c and conclude that some of the blue points in 2c represent the first days of the experiment (e.g. days 0 and 2). During this time, humidity was around 50 % according to 2a, but in 2c the x-axis only shows humidities above 87.5 %.
**A bottle that was filled on day 1 of the experiment stayed inside the ISCO for 23 days and experienced the full evolution of the relative humidity.  The values shown in Figure 2c represent the average relative humidity values experienced during the full storage duration of each bottle.**
**We will edit the caption of Figure 2 to better explain the aggregation of humidity values: "Isotopic change in water samples relative to the reference water as a function of _the mean relative humidity, which represents the average of relative humidity values during the full storage duration of each bottle_."**

Line 282: Please make clear that the "low relative humidity" refers to the chamber conditions (not the conditions inside the ISCO).
**We will clarify this by adding "... and low relative humidity _inside the ISCO autosampler_,..."**

Lines 296-298: If these temperatures and humidities are not shown anywhere, please indicate this to avoid confusion.
**We will edit this sentence: "Outside the ISCO autosamplers, air temperature (mean ± standard deviation) was 13.3±6.2°C in the outdoor setting and 18.6±4.7 °C indoors, while relative humidity was 73.5±23.0% outdoors with distinct daily fluctuations and 44.8±8.9% indoors _(values of temperature and relative humidity measured outside the ISCO autosamplers are not shown but are provided in the Supplement)_."**

Lines 312-314: If wind plays a role strongly depends on the geometry of the setup. A possible scenario that is not mentioned here so far is temperature-triggered gas volume changes. In contrast to Experiment 1 (climate chamber), Experiment 2 was characterized by significant daily temperature fluctuations (20°C and more inside the ISCO; outdoor setting; Fig. 4a). Upon heating (daytime), the air inside the ISCO expanded and some of the (moist) air was pushed out of the device. When temperatures dropped (at night), the opposite happened, i.e. the air in the ISCO contracted and sucked in fresh air from outside. This "sampler breathing" probably happened on a daily basis, resulting in a greater air exchange, which in turn (apparently) caused lower humidities inside the ISCO (and more evaporation). Maybe this effect could have played a role here as well.

**We will include in Line 313: „... is less important for causing isotope effects than the magnitude of the temperature fluctuations that may trigger gas volume exchanges.** *In contrast to Experiment 1 (ventilated chamber), Experiment 2 was characterized by significant daily temperature fluctuations (20 °C and more inside the ISCO; outdoor setting; Figure 4a). During the daytime when air temperature increased, the air inside the ISCO expanded and some of the (moist) air was pushed out of the device; when temperatures dropped (at night), the opposite happened, i.e. the air in the ISCO contracted and sucked in fresh air from outside. This "sampler breathing" probably happened on a daily basis, resulting in a greater air exchange with the outside, which in turn may have reduced the humidity inside the ISCO and resulted in stronger evaporation (Figure 4a). In addition,* **the greater potential for wind-driven ventilation of the outdoor ISCO** *may also have enhanced evaporative fractionation effects in the open sample bottles.***"**

Lines 328-329: This sentence is confusing. Do you mean ". . .exchange with the heavier RefA water. . ."?
**Correct, this was a typo. We will correct this in the revised manuscript.**

Lines 370-372: I am not sure if this statistical summary is sufficient. The mean value is close to zero, but I think you should at least mention that the isotopic shifts showed a fairly large scatter and ranged from about -2 (i.e. isotopic depletion) to about +3 ‰ (isotopic enrichment) in case of $\delta 2H$ (Fig. 7c and 7d). In case of $\delta 18O$, the maximum deviations are remarkable as well. Here, $\Delta \delta 18O$ scatters between about 0.7 and about +1.2 ‰ (Fig. S5c and S5d). Both values are clearly beyond the analytical precision.
**We will change the text to: "deviated *statistically* significantly from zero." We will also add a note to point out the large scatter in Figure 7 c,d: "***The isotopic differences of the samples in open bottles relative to the reference water scatter substantially with isotopic changes between -2 ‰ and +3.5 ‰, …***"**

Practical implications: Here, the wind issue is stressed again. Maybe the "sampler breathing" (see above) also deserves to be mentioned here. If it really played a role, a practical consequence may be the need of a thermal insulation of the ISCO (reducing the temperature fluctuations inside the device).
**We will change the respective sentence to "The larger isotopic change observed during the field deployment of Experiment 3 may be attributed to more variable climatic conditions (e.g., due to diurnal temperate variations)** *causing "sampler breathing",* **…". And we will also add "sampler breathing" at a later point in the practical implications Section: "Larger temperature and humidity contrasts due to diurnal fluctuations in outdoor conditions may have resulted in repeated evaporation and condensation inside the ISCO housing, and in enhanced vapor exchange between the sample bottles and the outside atmosphere** *("sampler breathing")***."**

Practical implications/Conclusions: The partly significant $\delta 2H$ and $\delta 18O$ values observed in Experiment 3 (see above) underscore the relevance of such "field controls". These pre-filled bottles (known isotopic composition and mass), placed into the automatic sampler upon field installation, allow for a hindsight evaluation of the samples' isotopic integrity. Although their advantage is somewhat obvious, it may be a good idea to explicitly recommend this technique to the reader.
**This is a good suggestion. We will add a sentence to the revised manuscript near the end of the "Practical implications" section: "***Control samples with known isotopic compositions in open, retrofitted and closed bottles placed in the autosampler for the entire storage duration, should be used to monitor composite isotope effects and allow a retrospective quality assessment of the automatically collected samples.***"**

**References**

Gröning, M., Lutz, H. O., Roller-Lutz, Z., Kralik, M., Gourcy, L., and Pöltenstein, L.: A simple rain collector preventing water re-evaporation dedicated for $\delta 18O$ and $\delta 2H$ analysis of cumulative precipitation samples, Journal of Hydrology, 448-449, 195-200, 2012.

Sonntag D.: Important New Values of the Physical Constants of 1986, Vapour Pressure Formulations based on the IST-90 and Psychrometer Formulae; Z. Meteorol., 70 (5), pp. 340-344, 1990.

Williams, M.R., Lartey, J.L., Sanders, L.L., 2018. Isotopic ($\delta 18O$ and $\delta 2H$) Integrity of Water Samples Collected and Stored by Automatic Samplers. Agricultural & Environmental Letters, 3(1), 1-5, 2018

---

## Referee Comment (RC2) · Anonymous Referee #2 · 9 Sep 2020

General comments

The manuscript entitled 'Evaluation of a low-cost evaporation protection method for portable water samplers' by von Freyberg et al. describes the development of a robust and inexpensive method for an evaporation reduction method for automatic water samplers that are often used in hydrology. In order to evaluate their developed setup, laboratory and field tests were conducted to simulate extremely dry and warm conditions, to test for vapor transfer between samples and to quantify the isotopic change during 3-week storage periods. It could be shown that the method efficiently protects the collected water samples from undergoing isotopic changes due to evaporative fractionation and vapor mixing and that the protection method significantly reduced isotopic fractionation over the 3-week periods under ambient climatic conditions in the field. The manuscript is well structured and nicely written. The topic of this promising approach fits well to the scope of the journal and appears to be of interest for isotope hydrologists. Most of my editing comments match those of Referee 1 and have already been addressed by the authors; therefore I only suggest minor revisions prior to acceptance and publication in Hydrology and Earth System Sciences.

Specific comments

Introduction, L. 45-47

I suggest mentioning styrofoam beads as an additional mechanical protection method, because this is commonly used as an evaporation protection method in ISCO-bottles.

L. 75, 81, 84

Please consider replacing 'Our. . .' by 'The. . .' at the beginning of these sentences, otherwise it sounds a bit like the conclusion section.

---

## Author Comment (AC1) · 10 Sep 2020

**Authors' response to the interactive comment by Anonymous Referee #2**
**on "Technical Note: Evaluation of a low-cost evaporation protection method for portable water samplers"**
**by Jana von Freyberg et al.**

**General comments**

The manuscript entitled 'Evaluation of a low-cost evaporation protection method for portable water samplers' by von Freyberg et al. describes the development of a robust and inexpensive method for an evaporation reduction method for automatic water samplers that are often used in hydrology. In order to evaluate their developed setup, laboratory and field tests were conducted to simulate extremely dry and warm conditions, to test for vapor transfer between samples and to quantify the isotopic change during 3-week storage periods. It could be shown that the method efficiently protects the collected water samples from undergoing isotopic changes due to evaporative fractionation and vapor mixing and that the protection method significantly reduced isotopic fractionation over the 3-week periods under ambient climatic conditions in the field. The manuscript is well structured and nicely written. The topic of this promising approach fits well to the scope of the journal and appears to be of interest for isotope hydrologists. Most of my editing comments match those of Referee 1 and have already been addressed by the authors; therefore I only suggest minor revisions prior to acceptance and publication in Hydrology and Earth System Sciences.

**We thank the anonymous referee for the positive evaluation of our manuscript. We address his/her two specific comments below (in blue bold font).**

**Specific comments**

Introduction, L. 45-47: I suggest mentioning styrofoam beads as an additional mechanical protection method, because this is commonly used as an evaporation protection method in ISCO-bottles.

**We will add the Styrofoam beads to the introduction: "Alternative mechanical evaporation protection modifications have been suggested, like *covering the water surface with Styrofoam beads (Angermann et al., 2017)* or placing a table tennis ball in the collection funnel…"**

L. 75, 81, 84: Please consider replacing 'Our…' by 'The…:' at the beginning of these sentences, otherwise it sounds a bit like the conclusion section.

**In the revised manuscript, we will change "our evaporation protection" to "*the* presented evaporation protection" (line 75), "our setup" to "*the described system*" (line 81, sentence changed from original based on comments from Referee #1), and "our design" to "*the presented* design" (line 81).**

**References:**
**Angermann, L., Jackisch, C., Allroggen, N., Sprenger, M., Zehe, E., Tronicke, J., Weiler, M., Blume, T., 2017. Form and function in hillslope hydrology: characterization of subsurface flow based on response observations. Hydrol. Earth Syst. Sci. 21, 3727–3748. http://dx.doi.org/10.5194/hess-21-3727-2017.**

---

## Author Response (AR1)

**Responses to Reviewers – Point-by-point response to reviewer comments on "Technical Note: Evaluation of a low-cost evaporation protection method for portable water samplers" by Jana von Freyberg et al.**

We would like to thank Nils Michelsen and one anonymous referee for reviewing our manuscript and for providing helpful comments. The point-by-point reply to the comments is given below. The comments provided by the reviewers are shown in regular font, and our responses in blue and bold.

**1) Authors' response to the interactive comment by Nils Michelsen (Referee)**

Dear colleagues,
With great interest, I have read your manuscript on the design of an evaporation reduction method for automatic water samplers, facilitating their use in isotope hydrology studies. Given the popularity of these samplers, the topic is, in my opinion, relevant for the HESS community. The simple concept is described in detail and since the required parts are low-cost and readily available, the method is easy to apply (if the reader has access to an automatic sampler). The experimental designs are well-thought-out and the corresponding results indicate that the suggested mechanism is indeed capable of reducing post-sampling evaporation, although there are limitations (certain climatic conditions, storage times).

The manuscript is also well structured and written.

I do have a few remarks (see below), but overall I think the manuscript should be published as a Technical Note in HESS, after minor revisions.

**We thank Nils Michelsen for this positive evaluation of your work and his comments, which were very helpful for improving our manuscript. We have addressed all comments in detail below (in blue bold font).**

Introduction: I think Williams et al. (2018) should be cited somewhere in the introduction. They have tackled the topic of post-sampling evaporation from automatic samplers before and should be given credit for their efforts. As their approach was rather different from yours (other evaporation reduction methods; experiments in insulated boxes instead of real automatic samplers), mentioning their work will provide additional context (and justification) for your work.
**Thank you. We will add this reference to the introduction.**

Line 47: That the tube dips into the collected water is obviously the most important aspect of this mechanism, but for sake of completeness, you could also mention the complementing aspect of the Gröning et al. (2012) design, i.e. the pressure equilibration tube (e.g. "Tube-dip-in-water collector with pressure equilibration", IAEA 2014). To additionally cite the pioneer work by IAEA (2002) is a good idea, but the reference is missing in the reference list.
**We will include IAEA (2002) in the reference list. We will also add a statement to the methods section:**
**"*Because a small gap remains between the syringe housing and the inner rim of the sampler bottle opening (i.e., not air-tight), pressure differences due to water flowing into the bottle will equilibrate with the outside conditions. Thus our system does not require an external tube for pressure equilibration such as the "tube-dip-in-water collector" proposed by Gröning et al. (2012).*"**

Lines 61-63: This sentence is a bit misleading. It suggests that only changes in air temperature and humidity would cause problems, but evaporation and vapor mixing would also occur if temperature and humidity remained constant.
**We will re-formulate the sentence: "However, because the sample bottles remain open during the sampling period, *vapor exchange may occur between the sample water and the atmosphere inside the sampler housing, which may alter* the isotopic compositions of the water samples in the bottles."**

Line 78: I suggest to refer to Figure 1a directly after "9 cm in length"
**We will add the reference to Figure 1a after the end of the suggested sentence and change the sentence to: "On the Luer tip, we fit a 1-mm inner diameter silicone tube approximately 9 cm in length, to reach the bottom of the sample bottle (Figure 1 *a*, b)."**

Line 80: The word "tightly" is misleading here. If the syringe housing is really plugged tightly into the bottle opening (i.e. air-tight), the whole mechanism would not work anymore. When additional water is supposed to flow into the bottle, air needs to be displaced and has to leave the bottle. You do mention this aspect later (Section 4, line 412-413), but this should be made clear here. You could even explicitly mention that you skip the "pressure equilibration tube" of the original concept (Gröning et al., 2012; IAEA 2002, 2014; see comment above).

**This is a good point. We will remove the term "*tightly*" from the sentence and clarify the setup by adding the explanation proposed above: "*Because there remains a small gap between the syringe housing and the inner rim of the sampler bottle opening (i.e., not air-tight), pressure differences due to water flowing into the bottle will equilibrate with the outside conditions. In this way, our system does not require an external tube for pressure equilibration such as the "tube-dip-in-water collector" proposed by Gröning et al. (2012).*"**

Lines 85-91: I appreciate that this potential pitfall is highlighted and that you provide a maximum filling rate. I can imagine that debris (e.g. sediment, insects) may further reduce the tubing diameter (1 mm is quite small) in a field setting. Hence, it is probably a good idea to use a screen at the intake (e.g. a funnel screen) and maybe you want to mention this explicitly somewhere.

**Good point. We will include a sentence after Line 86: "*In order to prevent debris (e.g. sediment, insects, leaves) from clogging the evaporation protection system, the streamwater intake or precipitation funnel can additionally be equipped with a screen.*"**

Line 104: For sake of consistency, this part should read ". . .during 62 two-to-three week cycles. . ." (as in lines 179 and 382)

**We will change this as suggested.**

Line 111: Please provide some details on the climate-controlled chamber. If this was a commercial chamber, please mention model and manufacturer. If it was a custom-made chamber, please indicate this.

**We will clarify this: "The ISCO autosampler was placed *on a heater inside a ventilated chamber* where the conditions were kept at approximately 35 °C air temperature and 11 % relative humidity."  We will also change the term "climate-controlled chamber" to "*ventilated chamber*" throughout the text (four instances in total).**

Line 113: Please also mention details on the temperature and humidity loggers (model, manufacturer, precision). Sometimes, it is these technical details that matter for the reader, particularly if they want to conduct similar experiments (as indirectly suggested in line 452). Knowing which logger can cope with such conditions (rel. humidity of up to 100 %), can be valuable.

**The specifications of the temperature and humidity loggers are: RHT30 humidity-temperature data logger; EXTECH Instruments, FLIR Commercial Systems Inc., Nashua, NH, USA; measurement accuracy ±1 % rel. humidity and 0.5 °C temperature).  We will include this information in the methods section of the revised manuscript.**

Line 217: Wouldn't $\Delta\delta^{18}O$ and $\Delta\delta^{2}H$ be more accurate?
**We decided to avoid double-delta notations for the sake of easier readability.**

Line 230: Please specify what the reference water is here (tightly sealed bottles).
**We will add this information to the sentence to be more specific: "…of the reference water (i.e., $\delta^{2}H_{reference}$, $\delta^{18}O_{reference}$ *of the water in the tightly sealed bottles*) for each sampling period."**

Lines 237-239: The 1°C temperature difference between the climate chamber and the ISCO (inside the climate chamber) is a bit surprising, particularly because the temperatures matched quite well before the logger as moved "to a more representative position".
**We will add a better explanation: "Air temperature outside the ISCO sampler was around 35 ± 1 °C with distinct diurnal variations (*a* 1.2 °C temperature drop at the beginning of the 4$^{th}$ day was caused by moving the *humidity-*temperature *logger from a position close to the heater* to a *higher* position *near the sampler's control unit to better represent the conditions* inside the *ventilated* chamber).**

Table 1: Probably, this should read ". . . relative to the reference water. . .".

**Correct, we will change this accordingly.**

Lines 264-267: The isotopic shifts observed for days 10 and 12 are indeed remarkable and I wonder if it is a coincidence that this was about the time when a humidity of about 100 % was reached. Could condensing water play a role here (i.e. liquid water drops flowing into the bottles)? Concerning the possible explanations that are given: I do not understand how spillage could explain the phenomenon.
**To check whether condensation might have played a role here, we calculated the dew point temperatures based on the Magnus formula and the parameter set from Sonntag (1990). As can be seen in the figure below, the dew point was reached inside the ISCO sampler after day 12 of the experiment. Even though this coincides with the day 100% relative humidity was reached, this would suggest that all samples filled on and after day 12 would also have been affected by condensation, which was not the case. In addition, because of our sampling design, all previous samples (days 0-10) were exposed to the dew point conditions as well but did not show such large isotopic differences as those samples from days 10 and 12.**
**We therefore can only speculate about the larger isotope differences in the samples of days 10 and 12. We removed the possible explanation of spilling and kept measurement errors during isotope analysis.**

[Figure]

Fig. 2: I am a bit confused about the x-axis in Fig. 2c. I was able to match all blue data points in 2b and 2c and conclude that some of the blue points in 2c represent the first days of the experiment (e.g. days 0 and 2). During this time, humidity was around 50 % according to 2a, but in 2c the x-axis only shows humidities above 87.5 %.
**A bottle that was filled on day 1 of the experiment stayed inside the ISCO for 23 days and experienced the full evolution of the relative humidity. The values shown in Figure 2c represent the average relative humidity values experienced during the full storage duration of each bottle.**
**We will edit the caption of Figure 2 to better explain the aggregation of humidity values: "Isotopic change in water samples relative to the reference water as a function of *the mean relative humidity, which represents the average of relative humidity values during the full storage duration of each bottle*."**

Line 282: Please make clear that the "low relative humidity" refers to the chamber conditions (not the conditions inside the ISCO).
**We will clarify this by adding "... and low relative humidity *inside the ISCO autosampler*,…"**

Lines 296-298: If these temperatures and humidities are not shown anywhere, please indicate this to avoid confusion.
**We will edit this sentence: "Outside the ISCO autosamplers, air temperature (mean ± standard deviation) was 13.3±6.2°C in the outdoor setting and 18.6±4.7 °C indoors, while relative humidity was 73.5±23.0% outdoors with distinct daily fluctuations and 44.8±8.9% indoors *(values of temperature and relative humidity measured outside the ISCO autosamplers are not shown but are provided in the Supplement)*."**

Lines 312-314: If wind plays a role strongly depends on the geometry of the setup. A possible scenario that is not mentioned here so far is temperature-triggered gas volume changes. In contrast to Experiment 1 (climate chamber), Experiment 2 was characterized by significant daily temperature fluctuations (20°C and more inside

the ISCO; outdoor setting; Fig. 4a). Upon heating (daytime), the air inside the ISCO expanded and some of the (moist) air was pushed out of the device. When temperatures dropped (at night), the opposite happened, i.e. the air in the ISCO contracted and sucked in fresh air from outside. This "sampler breathing" probably happened on a daily basis, resulting in a greater air exchange, which in turn (apparently) caused lower humidities inside the ISCO (and more evaporation). Maybe this effect could have played a role here as well.

**We will include in Line 313: „... is less important for causing isotope effects than the magnitude of the temperature fluctuations that may trigger gas volume exchanges. *In contrast to Experiment 1 (ventilated chamber), Experiment 2 was characterized by significant daily temperature fluctuations (20 °C and more inside the ISCO that was situated outdoors; Figure 4a). During the daytime when air temperature increased, the air inside the ISCO expanded and some of the (moist) air was pushed out of the device; when temperatures dropped (at night), the opposite happened, i.e. the air in the ISCO contracted and sucked in fresh air from outside. This "sampler breathing" probably happened on a daily basis, resulting in a greater air exchange with the outside, which in turn may have reduced the humidity inside the ISCO and resulted in stronger evaporation (Figure 4a). In addition,* the greater potential for wind-driven ventilation of the outdoor ISCO *may also have enhanced evaporative fractionation effects in the open sample bottles.*"**

Lines 328-329: This sentence is confusing. Do you mean ". . .exchange with the heavier RefA water. . ."?
**Correct, this was a typo. We will correct this in the revised manuscript.**

Lines 370-372: I am not sure if this statistical summary is sufficient. The mean value is close to zero, but I think you should at least mention that the isotopic shifts showed a fairly large scatter and ranged from about -2 (i.e. isotopic depletion) to about +3 ‰ (isotopic enrichment) in case of δ2H (Fig. 7c and 7d). In case of δ18O, the maximum deviations are remarkable as well. Here, Δδ18O scatters between about 0.7 and about +1.2 ‰ (Fig. S5c and S5d). Both values are clearly beyond the analytical precision.

**We will change the text to: "deviated *statistically* significantly from zero." We will also add a note to point out the large scatter in Figure 7 c,d: "*The isotopic differences of the samples in open bottles relative to the reference water exhibited substantial scatter, with values between -2 ‰ and +3.5 ‰, …*"**

Practical implications: Here, the wind issue is stressed again. Maybe the "sampler breathing" (see above) also deserves to be mentioned here. If it really played a role, a practical consequence may be the need of a thermal insulation of the ISCO (reducing the temperature fluctuations inside the device).

**We will change the respective sentence to "The larger isotopic change observed during the field deployment of Experiment 3 may be attributed to more variable climatic conditions (e.g., due to diurnal temperate variations) *causing "sampler breathing", …*". And we will also add "sampler breathing" at a later point in the practical implications Section: "Larger temperature and humidity contrasts due to diurnal fluctuations in outdoor conditions may have resulted in repeated evaporation and condensation inside the ISCO housing, and in enhanced vapor exchange between the sample bottles and the outside atmosphere *("sampler breathing")*."**

Practical implications/Conclusions: The partly significant δ2H and δ18O values observed in Experiment 3 (see above) underscore the relevance of such "field controls". These pre-filled bottles (known isotopic composition and mass), placed into the automatic sampler upon field installation, allow for a hindsight evaluation of the samples' isotopic integrity. Although their advantage is somewhat obvious, it may be a good idea to explicitly recommend this technique to the reader.

**This is a good suggestion. We will add a sentence to the revised manuscript near the end of the "Practical implications" section: "*Control samples with known isotopic compositions in open, retrofitted and closed bottles placed in the autosampler for the entire storage duration, should be used to monitor composite isotope effects and to allow a retrospective quality assessment of the automatically collected samples.*"**

L. 75, 81, 84: Please consider replacing 'Our…' by 'The….:' at the beginning of these sentences, otherwise it sounds a bit like the conclusion section.

**In the revised manuscript, we will change "our evaporation protection" to "*the* presented evaporation protection" (line 75), "our setup" to "*the described system*" (line 81, sentence changed from original based on comments from Referee #1), and "our design" to "*the presented* design" (line 81).**

**References:**

[revised manuscript text omitted]

---

## Author Response (AR2)

**Dear Christine,**

**Thank you for accepting the manuscript. The uploaded files contain the suggested corrections as follows (or responses are provided in bold blue font):**

Line 206 of the revised version: "... is the saturation vapor pressure (kPa) at the air temperature T (°C),...."; and then delete the definition of the air temperature in line 207

**Changed as suggested.**

Line 283 "humidities" and not "humilities"

**Changed as suggested, thank you for spotting this.**

Line 290: "very" can be deleted

**Changed as suggested.**

Even though $\Delta 2H$ was defined in the text, I would prefer to have $\Delta\delta 2H$ as it is the difference of isotope ratios given in the delta notation and not only the difference in deuterium (applies to entire manuscript and supplement)

**Changed as suggested in the entire manuscript (incl. figures and tables) as well as the supplement (text and data tables).**